# Light sheet fluorescence microscopy of cleared human eyes

Marie Darche [1,2], Ysé Borella [1,2], Anna Verschueren [1,2], Ivana Gantar [3], Stéphane Pagès[3], Laura Batti[3] & Michel Paques [1,2 ✉]

We provide here a procedure enabling light sheet fluorescence microscopy (LSFM) of entire human eyes after iDISCO + -based clearing (ClearEye) and immunolabeling. Demonstrated here in four eyes, post-processing of LSFM stacks enables three-dimensional (3D) navigation and customized display, including en face viewing of the fundus similarly to clinical imaging, with resolution of retinal capillaries. This method overcomes several limitations of traditional histology of the eyes. Tracing of spatially complex structures such as anterior ciliary vessels and Schlemm's canal was achieved. We conclude that LSFM of immunolabeled human eyes after iDISCO + -based clearing is a powerful tool for 3D histology of large human ocular samples, including entire eyes, which will be useful in both anatomopathology and in research.

[1] Paris Eye Imaging Group, 15-20 Hôpital National de la Vision, INSERM-DHOS Clinical Investigation Center, 1423 Paris, France. [2] Sorbonne Université, INSERM, CNRS, Institut de la Vision, Paris, France. [3] Wyss Center for Bio- and Neuroengineering, Geneva, Switzerland. ✉email: mpaques@15-20.fr

Clearing is a tissue processing technique that removes light-obstructing lipids to render tissues optically transparent. The recent development of light sheet fluorescence microscopy (LSFM) allows volumetric imaging of immunolabeled cleared samples and offers shorter acquisition time than confocal microscopy[1–4]. It also enables a more comprehensive analysis of relatively large and spatially complex structures, and much faster than complete reconstruction through traditional histology using tissue sections[5,6]. To date, many clearing protocols for clearing and labeling have been reported[7–9]. The most common techniques are CLARITY, BABB, 3DISCO, and CUBIC protocols[10].

Histopathology of an entire eye, in which neuronal, vascular, glial, pigmented, mesenchymal, and immune cells are intimately entangled between a collagenous shield and the vitreous gel, is notoriously difficult because of its heterogeneous composition[11]. Also, because the eye is spherical, 3D reconstructions from parallel sections or from flatmounts is cumbersome and prone to artifacts. Therefore, clearing would offer an elegant solution for histology of intact eyes. However, few papers have reported clearing of ocular tissues. Recently, we and others achieved LSFM of entire murine eyes (average diameter 3 mm)[12–16], enabling documentation of the time-course of vascular development, in particular the delicate organization of hyaloid vessels[17].

To date clearing of human eyes has been limited to some segments of the eye, such as the anterior segment[18] or the sclera[19]. Clearing of the entire human eye, a 6 cm³ sphere, has not yet been achieved. In addition, complete antibody penetration of the human eye has been difficult, as classic protocols only allow antibody penetration up to 1 or 2 mm in depth[20]. Here we report the achievement of LSFM of entire immunolabelled human eyes by combining a modified iDISCO+ clearing protocol, refined procedures for bleaching, and improved antibody penetration using linearization. Acquisition of images through a custom-built mesoSPIM[21] followed by extensive numerical dissection enabled multiscale, customized observation of the tissues forming the eye, including pseudo-fundoscopic (i.e., en face) viewing.

## Results

Four entire eyes and ten anterior segment samples from donors aged 16 to 94 years underwent clearing and LSFM (Schematic in Fig. 1a). Figure 1c shows a macroscopic view of a human eye after clearing in the imaging cuvette. Figure 1d–k and Supplementary Video show the imaging results from a cleared eye of an 84-year-old woman (a schematic of the structures of the human eye is shown in Fig. 1b). The fluorescence signal, labeling ColIV and SMA, showed homogeneous penetration of antibodies and light throughout the sample (Fig. 1d,e). Virtual dissection allowed isolation of the various ocular structures. The entire vascularization of the retina could be visualized similarly to en face clinical imaging ("pseudo-fundoscopy"; Fig. 1f and h). The complex vascular organization of the choroid could be observed in detail in the samples (Fig. 1g). The foveal avascular zone (Fig. 1i) and the multilayered retinal and choroidal circulations (Fig. 1j–k) were clearly defined.

Representative examples of LSFM of anterior segment samples are shown in Fig. 2 and Supplementary Video. Their smaller volume (typically within 12 × 12 × 5 mm) enabled them to be examined at higher resolution, because higher resolution objectives with smaller working distances could be used. Optical sections through the entire sample stained with anti-tubulin III (Fig. 2a) allowed the visualization of the innervation of the anterior segment. En face section of the anti-tubulin III labeled cornea of the anterior segment of a 16-year-old woman (Fig. 2b) showed the physiological organization of corneal nerves; besides, an 89-year-old donor (Fig. 2c) showed disorganized axons,

comprising foci of tortuous axons, presumably microneuromas as clinically described in aging patients using in vivo confocal microscopy. In sagittal sections, details of the innervation of the limbus, ciliary processes and iris are shown in Fig. 2d.

The Schlemm's canal (Fig. 2e, f, green circle) and downstream aqueous veins (Fig. 2e, yellow arrowheads) could be traced and rendered in 3D (Fig. 2g).

Double labeling of neurovascular networks (Fig. 2h) enabled to visualize the high density and details of nerves in the iris and sphincter muscle (Fig. 2i) as well as the pattern of the ciliary bodies and vessels in the sclera (Fig. 2j). A long anterior ciliary artery passing through the sclera and ciliary muscles joining the major arterial circle of the iris is shown in Fig. 2k.

## Discussion

Here we achieved labeling, clearing and imaging of entire human eyes, allowing visualization of the different 3D structures of the eye in intact samples. While several protocols have addressed the challenge of the mouse eye in recent years, to the best of our knowledge none has proved able to image entire human eyes at a cellular resolution. Differences between the mouse and human eye can likely explain the lack of 3D imaging of an intact human eye before now.

In order to develop our ClearEye protocol, each step was designed to overcome the unique biological challenges of the human eye (Supplementary Fig. 1a). The human eye has a high concentration of melanin and requires specific processes for bleaching. To overcome this, we used a combination of photo-bleaching, a technique used in histopathological protocols for highly pigmented samples of melanoma[22], and oxygen peroxide ($H_2O_2$)[23], most recently used in clearing of the mouse eye[16]. Next, to improve penetration of antibodies through the vitreous and sclera, we used an immunolabeling protocol based on the modulation of antibody-antigen binding[24,25]. SDS is an anionic detergent which coats proteins in negative charges, leading to partial loss of their tertiary structure and producing a more linear protein molecule. Considering the link between antibody tertiary structure and antibody-antigen binding, it has been shown that concentrations of SDS over 0.01% leads to an inhibition of immunobinding;[26] This property has been used in the SWITCH protocol[25]. Indeed, antibodies in contact with SDS will lose their antigen-binding properties, allowing them time to homogenously penetrate deeper into tissues. Washing the SDS after the initial incubation rescues the antigen-binding properties, allowing for a more homogeneous staining of deep tissues, less affected by the often observed gradient of stronger labeling on the outside of the sample. This also solved the problem of the vitreous, whose meshwork tends to block penetration of antibodies, an issue frequently encountered in flat mounted retinas.

The final hurdle in whole eye clearing and imaging is choosing a microscope able to image 3 × 3 cm samples in a reasonable amount of time, which we found in mesoSPIM. When compared to other techniques such as confocal microscopy, the LSFM is 2–3 orders of magnitude faster than point-scanning methods[3]. The mesoSPIM is a custom-built microscope based on the axional scanned light-sheet microscopes (ALSM) technology[27] enabling the examination of centimetric samples.

Clinical examination of the fundus of the eye is done by both en face imaging (e.g., funduscopy) and transverse imaging (OCT). The current gap between histology and in vivo imaging arises from the fact that 3D reconstruction and en face viewing using conventional sectional histology is cumbersome and by essence prone to reconstruction artifacts inherent to realignment, and conventional confocal microscopy does not allow to examine intact eyes. On the other hand, 3D visualization of intact eyes allows direct comparison

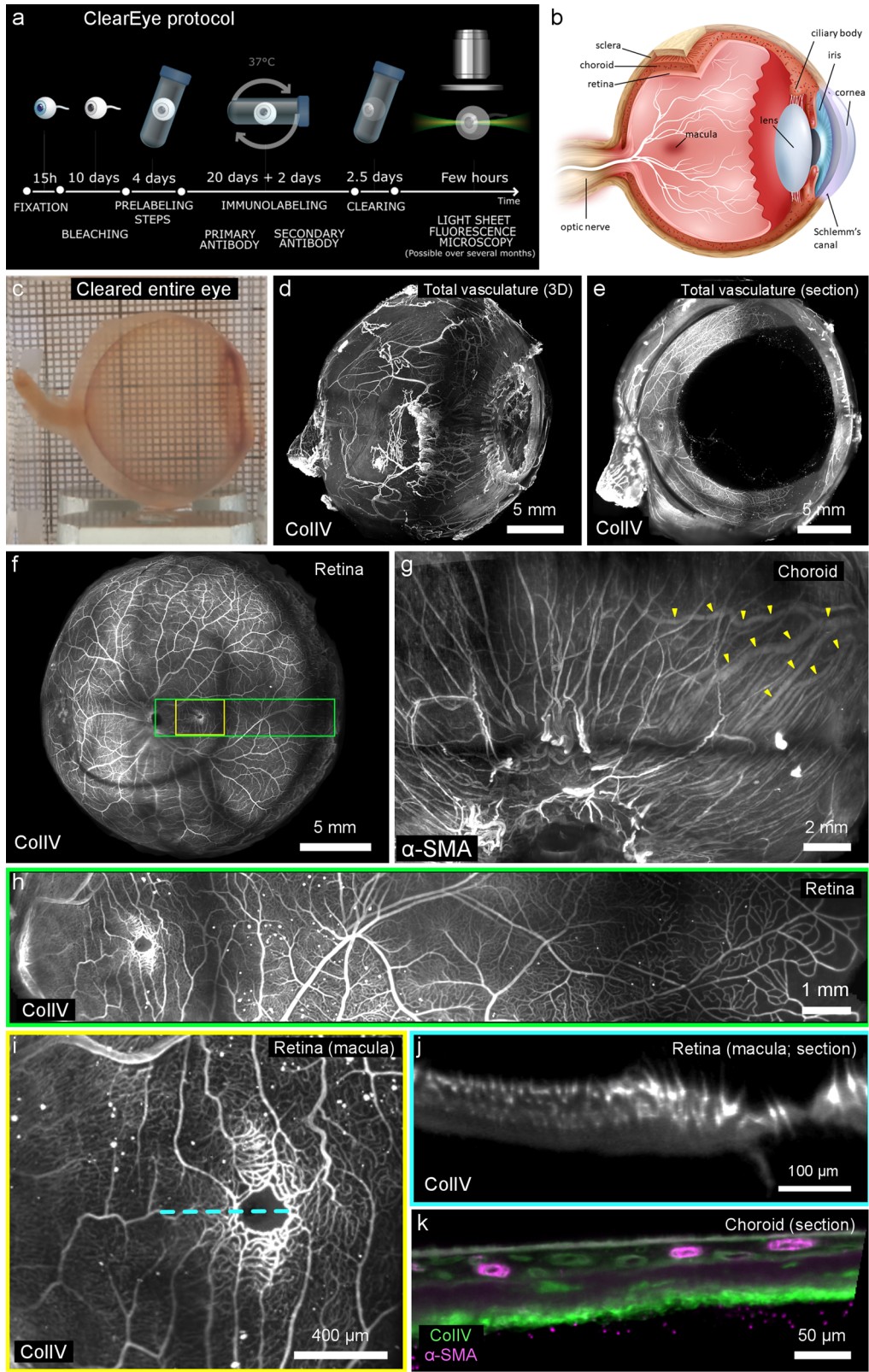

**Fig. 1 Overview of ClearEye protocol and LSFM of an entire human eye after ClearEye protocol. a** Graphic summary of ClearEye protocol. **b** Schematic representation of ocular structures. **c** Macroscopic aspect of a human ocular globe post-ClearEye procedure. All subsequent (**d**–**k**) are from the same eye of an 84 year-old female. All panels show anti-ColIV immunolabeling except (**g** and **k**), which show anti-alpha SMA labeling. **d** Visualization of all vascular networks. **e** Sagittal optical section. FfPseudo-fundoscopic (en face) viewing of retinal vessels. **g** en face view of choroidal vessels labeled using an anti-alpha SMA antibody, with the draining vortex vein highlighted with yellow arrowheads. **h** Magnification of the area indicated by the green square in (**f**), showing the vessels from the optic nerve (on the left) to the anterior retina (ora serrata, on the right). **i** Magnification of the area indicated by the yellow square in (**f**), showing perifoveal capillaries. **j** Optical section along the blue dotted line in (**i**) showing the microvascular layers of the retina. **k** Optical section of the choroid showing strong anti-alpha SMA labeling of choroidal arteries (See also Supplementary Video). (ColIV: collagen IV, αSMA: alpha-smooth muscle actin).

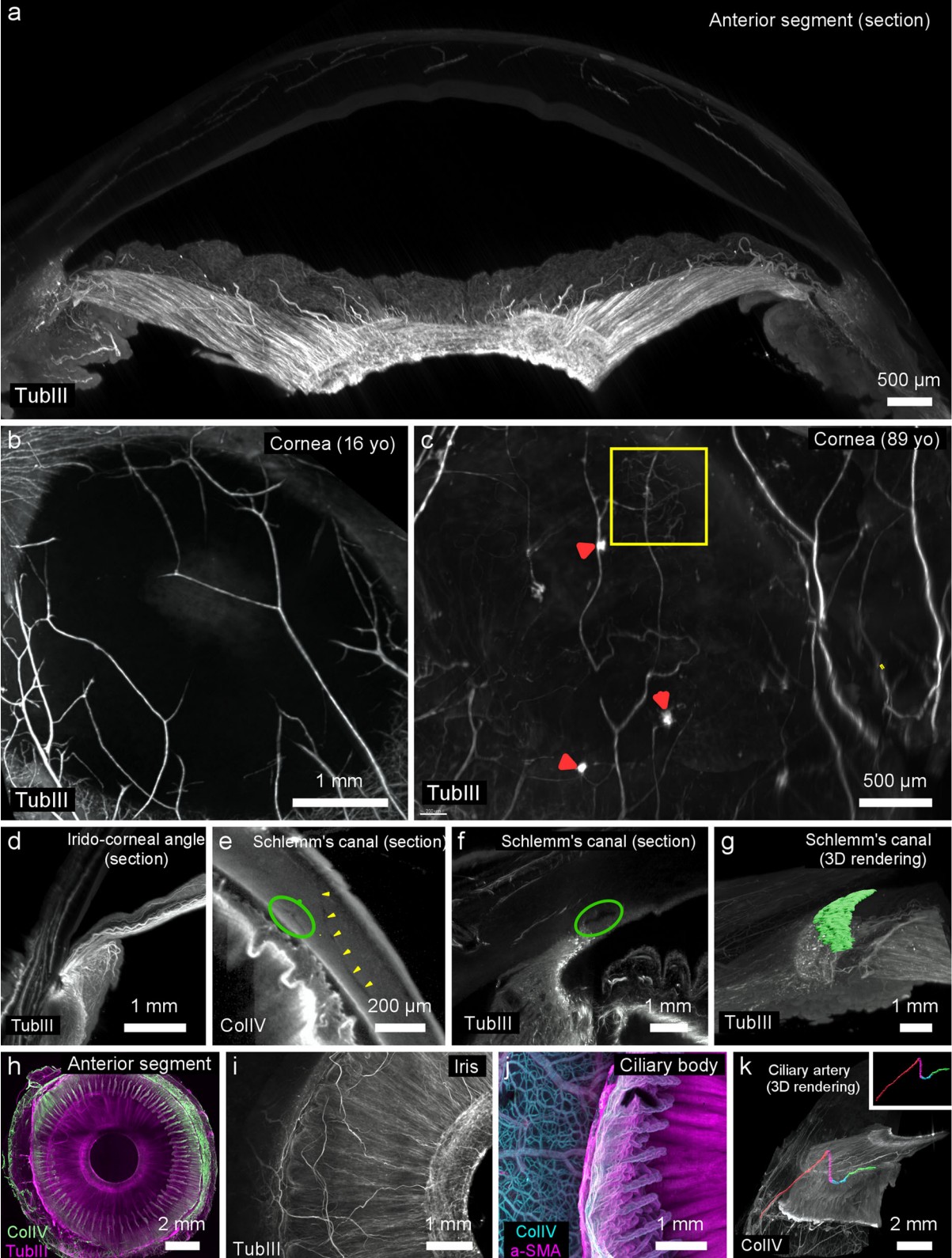

**Fig. 2 LSFM of the anterior segment after the ClearEye procedure.** Images are of anterior segments from six patients (age range 16–89 years). **a–d**, **f**, **g** and **i** Show anti-TubIII immunolabeling. **e** and **k** Show ColIV labeling. **h** Shows anti-ColIV and TubIII labeling. **j** Shows anti-ColIV and anti-SMA labeling. **a** Anterior segment section showing innervation of iris, cornea, and ciliary body. **b** En face visualization of corneal innervation in a 16-year-old donor. **c** Corneal innervation in an 89-year-old donor with dry eye showing microneuromas (red arrowheads) and tortuous nerves (yellow square). **d** Innervation of the ciliary body, iris, and cornea. **e**, **f** and **g** Scan of a collector vein (**e**, yellow arrowheads) downstream of the Schlemm's canal (oval in **e** and **f**) with manual segmentation and 3D rendering (**g**) (see also Supplementary Video). **h** Double labeling of vessels (ColIV, green) and nerves (TubIII, magenta) in the anterior segment. **i** Innervation of the iris. **j** Close up of the ciliary processes and scleral vascularization (on the left). **k** Tracing of the pathway of a long anterior ciliary artery from outside the sclera (red), through the sclera (magenta), into the ciliary muscles (blue), and ending in the major arterial circle of the iris (green). (TubIII: Tubulin-III; ColIV: collagen IV).

to clinical imaging. Our procedure indeed can achieve a "pseudo-fundoscopic viewing" of the fundus, as we show for the foveal avascular zone (Fig. 1f and h). LSFM of cleared eyes is therefore more convenient for the exploration of the phenotype of human diseased samples. This was illustrated by the fact that in LFSM stacks of cleared samples of the anterior segment we could identify microneuromas (Fig. 2c), which were recently described by in vivo confocal microscopy and are presumed to be related to dry eye syndrome and corneal pain[28,29]; to the best of our knowledge these have not yet been observed histologically. The Schlemm's canal is a circular canal playing a crucial role in the aqueous humor outflow from the eye. It is difficult to analyze in histology because of the paucity of cellular or molecular targets. After ClearEye, Schlemm's canal and downstream aqueous veins were segmented and rendered in 3D (Fig. 2g). The detailed volumetry of Schlemm's canal is of particular interest for the study of glaucoma, because understanding its size and shape in disease may improve our understanding of its impedance, resistant to outflow, and dysfunction during disease[30,31].

Some features of the ClearEye protocol can be improved in future implementations. Artifactual detachment of the retina or spots of oxidized pigments in the subretinal space were observed sporadically in some samples (see Supplementary Fig. 2a). Also, antibody penetration in the optic nerve is suboptimal, although in the future this may be improved by ablation of meningeas. Another important consideration for large samples is that the image quality can depend on the volume of the sample. This is due to both the total size of the sample and the moderate inhomogeneity of refractive index in some structures. These constraints were mitigated by using dual side illumination to increase the penetration of the light through the whole sample and by correcting the misalignment of illumination with the focal plane throughout the whole specimen thickness, allowing us to achieve good quality images, with a resolution between 5–8 um axially and 1.4–8 um laterally. Currently, image processing is done manually and is time-consuming. This could be surpassed by the development of dedicated analysis tools.

In summary, we here demonstrate the feasibility of LFSM of cleared entire human eyes for nondestructive examination of human eyes. Further, cleared eyes remain examinable by LFSM for several months when stored correctly, allowing repeat examinations of samples. This paves the way for a comprehensive, customized, and quantitative three-dimensional exploration of human ocular structures. The possibility to manipulate stacks in 3D with minimal loss of resolution is particularly exciting. This will greatly facilitate translational research, since in vivo clinical and postmortem examinations of both the anterior and posterior segments can be compared more directly. *In toto* LFSM of entire human eyes is therefore of interest for scientific as well as medical purposes[32]. Several applications in ophthalmology can be envisioned, as comprehensive documentation of ocular tissues will facilitate the mapping of focal abnormalities and the analysis of spatially complex and disease-relevant structures. *In toto* LFSM may help to explore the structural consequences of complex diseases involving several ocular structures, such as age-related macular degeneration, uveitis or myopia[33,34]; in tumors, *in toto* examination would facilitate the detection of foci of local dissemination of cancer cells. Mapping of protein deposits in eyes from patients with Alzheimer disease may also be facilitated by this approach. Altogether, clearing and *in toto* imaging of human eyes is a powerful tool with wide ranging applicability in medical and basic research.

## Methods

**Eye samples**. Eye samples from donors were obtained through the Lion's Gift of Sight (Saint Paul Minnesota, USA), operating under the regulations of the U.S. Food and Drug Administration (FDA) and the Eye Bank Association of America. Consent from the donor was obtained and the next of kin received no charge or monetary gain from the donation. Harvesting of the samples was done <24 h post-mortem. The lens was extracted through a corneal incision and the samples were fixed in 4% paraformaldehyde (15710, ThermoFisher scientific) in PBS overnight at room temperature. The lens was removed because we often observed reopacification of cleared lenses over time. Importation in France was done under relevant regulations for transfer of human tissues (CODECOH DC-2015-2400). All ethical regulations relevant to human research participants were followed.

**ClearEye protocol**. We developed a clearing protocol termed ClearEye (graphically depicted in Fig. 1a) for intact human eyes (major ocular structures schematized in Fig. 1b). The timing of the successive steps of the ClearEye procedure are schematized in Fig. 1a. Whenever a shaker at 37° is mentioned, the apparatus used was an Incu-shaker mini (Benchmark scientific); at room temperature, the apparatus was a Ms Major Science Rocking Shaker, on a low rocking setting (10–15 RPM). Samples were dehydrated in successive baths of phosphate buffered saline (PBS)/methanol (322415, Sigma Aldrich) (50%, 80%, and 100% methanol, 2 h each). Bleaching was done with 11% hydrogen peroxide (23613.297, vWR) in methanol, under white light (A025416, Manutan, 11 W) for 10 days with agitation.

*Bleaching*. To help with pigment elimination, the bleaching solution was renewed every day and a gentle flow toward the interior of the eye was generated using a pipette during each bleaching solution renewal. After bleaching, rehydration in PBS was achieved using successive baths (100% Methanol, 80%, 50%, 0% Methanol/PBS, 2 h each). Four to six iterative 90 min PBS washes were done over 1 day.

*Labeling*. To improve penetration of antibodies, we used an immunolabeling protocol derived from the System Wide control of Interaction Time and Kinetics of CHemicals (SWITCH) protocol[25], which achieves deeper penetration of antibodies by modulation of antibody-antigen binding using sodium dodecyl sulfate (SDS). Samples were first permeabilized and equilibrated for 4 days in PBSGT-Sw (PBS 1X containing 0.2% gelatin: 24350262, Prolabo, 5% Triton X-100: X100–500ml, Merck Millipore Sigma-Aldrich, and 10 mM SDS: L3771, Sigma Aldrich) under gentle agitation at 37 °C. We used the following primary antibodies: goat anti-Collagen IV (diluted 1/1000, 134001 Biorad), mouse anti-alpha-smooth muscle cell actin (SMA; diluted 1/2000, A2547 Merck Millipore Sigma Aldrich), and rabbit anti-Tubulin III (diluted 1/2000, T2200-200uL Merck). Antibodies were first incubated separately from samples in PBSGT-Sw for 2 h. Samples were then incubated in this primary antibody solution for 20 days at 37 °C, with agitation of 70 RPM. After 20 days, samples were rinsed and left for 4 days in PBSGT without SDS, to allow primary antibodies to bind to their antigenic targets. After filtering the secondary antibodies mix using 0.22 μm Syringe Filter Rotilabo® (ROTH- KH54.1), secondary antibodies (cross-absorbed donkey secondary antibodies conjugated to AlexaFluors from ThermoFisher, diluted 1/500) were then incubated in darkness for 48 h in PBSGT at 37 °C with 70 RPM agitation, and washed three times for 2 h in PBS. To mitigate tissue softening and damage during clearing, samples were fixed again in 4% paraformaldehyde for 45 min.

*Clearing*. For clearing, samples were dehydrated in successive PBS/methanol baths (20% methanol overnight, then 40%, 60%,

80%, 2 × 100%, each 2 h), then put overnight in a 2/3 dichloromethane (DCM) (270997-1 L, Merck) 1/3 methanol solution. Samples were then immersed in 100% DCM for 45 min, and then in BenZyl Ether (DBE) (108014, Merck). DBE baths (2 h) were changed until no swirls were seen in the solution following mild agitation of the tube. Cleared samples were then stored in DBE in darkness, where transparency and immunofluorescence were stable for at least a year.

**Imaging**. For imaging, cleared human eyes (macroscopic views shown in Fig. 1c) were placed in an immersion quartz cuvette (3 cm × 3 cm × 4 cm) filled with the imaging medium DBE. Imaging was performed using a custom made mesoscale selective plane-illumination microscope mesoSPIM[21] system at the Wyss Center in Geneva, which allowed imaging of a human eye in its entirety (travel range 44 × 44 × 100 mm) and provided near-isotropic resolution 3D datasets. Briefly, the sample is illuminated by one of the two digitally scanned light sheets coming from opposite directions. The excitation paths also contain galvo scanners for light-sheet generation and reduction of shadow artifacts due to absorption of the light-sheet. In addition, the beam waist is scanned using electrically tunable lenses synchronized with the rolling shutter of the sCMOS camera. This axially scanned light-sheet mode (ASLM) leads to a uniform axial resolution across the field-of-view (FOV). Emitted fluorescence is collected by Olympus MVX-10 zoom microscope with a 1× objective (Olympus MVPLAPO 1×) and imaged on a digital camera (Hamamatsu ORCA-Flash 4.0) at 5 Frame Per Second (FPS). Excitation wavelength of the autofluorescence (Supplementary Fig. 2b), 561 and 647 nm labels were set at 488, 561 and 647 nm, respectively with an emission 530/40 nm bandpass filter, 593/40 bandpass and 663 LP filter respectively (BrightLine HC, AHF). All acquisitions were made from only one orientation of the human eye in a multi tile setup (2 × 2) using one of the two digitally scanned light sheets. A separate dual side illumination was used to increase image quality and the penetration of the light through the whole sample. To mitigate the constraints of imaging a large size sample, sequences of z-sub-stacks were therefore taken at different focal ranges. Z-stacks were acquired with a zoom set at 0.8X at 5 µm spacing, resulting in an in-plane pixel size of 8.2 × 8.2 µm (2048 × 2048 pixels). This resulted in an undersampled 3D image according to the Nyquist sampling rule. Indeed, the effective detection NA of the MVX-10/MVPLAPO varies with zoom and can be estimated as NA = 0.065 at 0.8X. Based on this value, the diffraction-limited lateral resolution would be 3.8 µm.

**Data analysis**. After data acquisition z-sub-stacks were concatenated using Image J and the tiles were stitched in 3D using the Grid Collection Stitching plugin tool in TeraStitcher (BMC Bioinformatics, Italy). The resulting HdF5 files were then converted to .ims format and analyzed using the ImarisBitplane software or the virtual reality software Syglass. Imaris software (versions 9.5–9.7, Bitplane) was used to render and study samples in 3D. To isolate target ocular structures and vascular beds, built-in tools of slicers or manual segmentation and 3D rendering are used (see Supplementary Fig. 1b). Built-in Imaris animation and snapshot tools were used for video and image acquisitions. FIJI's plugin ScientiFig was used to create scientific figures and iMovie© (v10.1.11) was used for video editing.

Supplementary Table details the labeling and imaging characteristics for all figures.

**Reporting summary**. Further information on research design is available in the Nature Portfolio Reporting Summary linked to this article.

## Data availability
The histological images supporting Fig. 1 and Fig. 2 are publicly available in the figshare repository, at the following addresses : https://doi.org/10.6084/m9.figshare.24162243.v2[36] and https://doi.org/10.6084/m9.figshare.24162237.v2[35]. Any remaining information can obtained from the corresponding author upon reasonable request.

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

## Acknowledgements
Kaitryn Ronning for the english revisions and useful advice. Funded by the Institut Hospitalo-Universitaire FOReSIGHT (ANR-18-IAHU-01), Region Ile-de-France (EX047007 - SESAME 2019) and Association Contre l'OVR. The funding organizations had no role in the design or conduct of this research.

## Author contributions
M.D. and M.P. conceived and initiated the project. M.D. and M.P. designed research; M.D. performed research; I.G., S.P., L.B., and M.D. imaged the samples and processed the raw data. M.D. segmented and analyzed data; M.D., Y.B., A.V., and M.P. wrote the paper. Y.B. and M.D. created the figures and the Fig. 1 schematics.

## Competing interests
The authors declare no competing interests.
