## [Peer Review File · Communications Biology]

Reviewers' comments:

Reviewer #1 (Remarks to the Author):

In this manuscript, the authors describe the first full imaging of cleared human eyes using light sheet fluorescence microscopy and a customised iDISCO clearing protocol. The manuscript is clear and well written, with impressive results. I particularly enjoyed the supplemental movie, which presented an impressive demonstration of the main results.

However, I miss a fuller description of the imaging performance and comparison to other techniques, such as confocal microscopy. How much faster/easier is the light sheet imaging? Are there regions of the sample that are less well resolved than others? What is the typical achieved resolution? How much is the axial resolution improved by using the ASLM mode compared with a thicker, longer sheet? Is it possible to obtain a scattering-only volume (cf. reflection confocal microscopy) to see all structures within the eye, not just those labelled with antibodies? Is there a fundamental limitation that explains why only two-colour volumes are presented?

Furthermore, in Figure 1 panel C, the cleared eye has a distinct pink hue. Can the authors identify which residual component is responsible for this? It would appear that this may cause issues with illumination absorption near the outer boundary of the iris. Is this the case?

The authors note that they remove the lens as they often observed reopacification of cleared lenses over time. Is any reopacification of other eye tissues observed?

In the methods section, the authors state "this axially scanned light-sheet mode (ASLM) leads to a uniform axial resolution across the field-of-view (FOV) of 5 μm ", but also "Z-stacks were acquired with a zoom set at 0.8X at 5 μm spacing". Hence, the Nyquist sampling criterion is not satisfied and so the true maximum possible axial resolution is 10 μm . Similarly, with an XY pixel pitch of 8.2 μm , this corresponds to a maximum possible resolution of 16.4 μm . What resolution is actually achieved in practice, and how does this vary over the sample?

If these queries can be addressed, I would strongly recommend publication in Communications Biology.

Minor issues:

1. In the abstract, "light sheet fluorescence microscopy" is given the abbreviation "LFSM" which is mis-ordered. This choice is consistent throughout the manuscript, but would be less confusing if changed to the common LSFM.
2. Figure 1 panel A misspells "fluorescence" as "fluorescente".
3. In the same panel, the timeline suggests that the light sheet imaging takes several months, which I do not think is the case (I believe the suggestion is supposed to be that the imaging can be started at any point within a window of several months before sample degradation). Perhaps the figure can be altered to make this less confusing.

Reviewer #2 (Remarks to the Author):

Authors Darche et al. report on a new method of optically clearing the human eye for light sheet fluorescence microscopy. The authors have demonstrated the advantage of this application in its

ability to successfully visualise the ocular structures of the human eye in its native configuration to greater depth and higher resolution. This is an important area of study. However, there are several revisions that are recommended to strengthen the paper and make it accessible to a larger audience as it has the capacity to be of value to the wider field, outside of vision sciences.

Introduction:

1. Line 20 -23: references?
2. The introduction requires a more comprehensive literature review on the clearing and imaging methods that have been previously conducted on the human eye to date? If it is only very few, then there is space to indicate each one. In addition, what specifically makes this method better and the rationale for its use over others? This requires greater clarity in introduction. A search will demonstrate that other methods have been published (PMID: 28574496, PMID: 35859136). Together, this will help to support your statement in Line 32 – 33.
3. Line 35 – explain linearization.
4. Line 42- 43 – clarify what you mean by ‘modulation’ of antibody-antigen binding?

Results:

5. Line 51 - 52 – not clear on how imaging of ocular vascular bed is achieved? Was vasculature stained with CD31?
6. Line 44 – 53. The description of the figures in the results require more detail and thought. Although, the figure legends do have detail within them, reference to the figures in the results should explain what procure has been done and the appropriate structures that are being shown. Eg, what is the yellow arrows pointing to, scale bar on K?
7. Line 51 – fig 1, F, G should be Fig F, G, I – please carefully check for formatting, grammatical errors and typos throughout
8. Upon immediate glance, the figures are not intuitive to what is being shown other than a assumption that it is blood vessels. To help the reader, it would be useful to have a small subtitle on each frame (Eg. Panel A – could have the title ClearEye Method), and if possible, to pseudo-colour the appropriate panels as was done in Figure K. Eg. It is not evident that Panel G vs H is of the vasculature of the retina and choroid, respectively.
9. Figure 1, Panel B – instead of number of the structures, can you annotate directly onto the eye schematic.
10. Same comments for Figure 2 as made for Figure 1
11. Line 70 – can anti-tubulin III be pseudo-coloured?
12. Line 72 – Fig 2C – what are yellow arrows pointing to.
13. Line 77 – explain why segmentation and 3D rendering is useful for the Schlemm’s canal

Discussion

14. Similar to comments made about introduction, here it would be useful to compare with other clearing and imaging systems that have been used and/or how the lack of such methodology as the one your have established has hampered the ability to fully visualise the delicate nerves, blood vessels and immune cells found within the eye.

Methods:

Given this this is a methods-based study, the following details will be useful for reproducibility of this work.

15. Line 151 – 153 – This begs the question of how important it is to fix tissues in 4% PFA O/N? Have you tried with different fixation times?
16. Line 153 – fixed O/N at room temperature or 4 degrees C?
17. Line 154 – clarify whether removal of lens post-fixation doesn’t lead to collapse of the eye.
18. Line 159 – did the eyes have to be injected with hydrogen peroxide or were the eyes only immersed in bleaching solution?
19. Line 162 – approximately how many PBS washes across the day and how long was each?
20. Line 166 – do you mean antibody-antigen binding?
21. Line 166 – what specifically is SDS doing, explain? Does it act like antigen-retrieval to enhance epitope binding?
22. Line 172 – antibodies incubated in PBSGT-Sw for several hours – how many hours is that?

23. Line 180 – 182 – explain why you decided to use DCM and DBE? Are these used in other published clearing papers? If yes, reference.
24. Line 183 – how long and under what conditions can cleared samples be stored in DBE for? Is this the final storage medium?
25. Line 185 – what is the imaging medium?

Overall, I highly commend the authors on undertaking the series of experiments and obtaining the subsequent results that have led to the development of a new clearing and imaging method of the human eye. In particular, the ability to visualise the anterior segments of the eye, including the innervations is significant. There has no doubt that this project has required many optimisations and has met with numerous challenges and redirections. However, the persistence of this project to generate this data has enabled an important body of work to come to light, and become available to the scientific field. Therefore, the findings from this manuscript will be well received by the vision sciences, ocular immunology, neuroscience and other imaging fields. Well done.

Reviewer #3 (Remarks to the Author):

Brief summary of the manuscript

=====

This manuscript is a brief communication that, for the first time, proves the utility of sample clearing and light sheet fluorescence microscopy for imaging intact whole human eyes. The authors have developed a protocol for clearing and imaging human eyes, ClearEye, based on their previous experience working with mouse eyes. The authors briefly discuss the relevance of the resulting images and analysis techniques for clinical research.

Overall impression of the work

=====

The manuscript is technically sound using adaptations to well-known techniques and the concept is a logical extension from previous methods of clearing mouse eyes, <https://www.nature.com/articles/s42003-022-04104-2>. The work is convincing, however there could be further effort to describe the methodology and analysis to aid readers in repeating this work. Additionally, refinement of the technique to be reliable would be beneficial to the community, or at least highlighting the critical steps that are unreliable.

This paper will influence thinking in the field, as it opens clinical researchers to the option of imaging intact whole human eyes. The exciting features of this work are the novel images of whole human eyes at a decent resolution (8 μm), the outstanding ease with which many structures in the eye can be identified, and the data manipulation possible using virtual dissection. The workflow is not necessarily relevant to those outside of the field, as it is specific to whole human eyes.

No other experiments would strengthen this paper in my opinion, the experiments conducted are of utility on their own and appear novel. The methodology is completely manual, no new tools for virtual dissection are used or created. There appears to be no major biological claims from the paper, so this is purely a technical paper, using examples of the biology of the eye.

I congratulate the authors on their efforts!

Specific comments, with recommendations for addressing each comment

=====
=====
Main comments:

1. As one of the main outcomes of the paper is the description of the clearing protocol, it should be described in more detail by:
 - Including an extra figure (main or supplemental) to describe the sample processing steps in more detail (images of the setup, apparatus etc.), expand on methods section slightly
 - Including an extra figure (main or supplemental) to describe the analysis conducted, expand on methods
 - If this is an unfinished protocol with issues (see line 106), it may be best to make the refinements before publication

2. Further discussion of the context and benefit of the technique:
 - Over existing techniques
 - For clinicians or ophthalmology research in general

Minor comments:

3. Grammatical error corrections to improve readability
4. Copy-editing would benefit this manuscript to improve grammar and therefore readability as Nature Communications Biology does not undertake copy-editing for authors, examples include L16 grammar, L23 grammar, L31/32 repetition, L46/47, L55, L57, L58, L78, L97, L100-102, L111, L173
5. Headings for the text sections would be highly beneficial to allow the reader to navigate the paper
6. The development of an analysis tool specific to data from the eye would be beneficial and should be discussed, as currently it is done by hand with non-specific tools
7. Are there any novel biological findings that can be highlighted?
8. Discuss the comparison of this protocol to the existing best methods (sectioning and mounting) using a figure, discuss or show resolution differences and discuss what can be resolved using this technique over others

Comments by page/line:

Page 1

Line 13 - grammar: should LFSM be LFSM for Light sheet fluorescence microscopy? Written throughout paper as LFSM

Page 2

L20-23 - further information: is there a reference for this claim?

L29 - inclusion of this reference may be beneficial here:

<https://www.ncbi.nlm.nih.gov/pmc/articles/PMC8374524/>

L35 - further information: could you elaborate on linearisation further somewhere in the methods or text as it is not clear from the references what this means

L36 - clarification: what is 'extensive numerical dissection', does this mean 'analysis'? A more informative explanation for the analysis could be used here

L39 - further information: if this is an adaptation of the mouse eye clearing protocol, a comparison to that protocol would be beneficial here to highlight what was changed and why the human eye required the adaptation (the current description is not specific to the 'human' eye)

L39 - further information: the steps taken to develop the protocol are not shown and may be

beneficial to help the community understand decisions made when clearing different tissues. Include data from the development of the protocol if possible? Was a reference used to assist your choices, or was it trial and error? Some elaboration of the clearing protocol in the main results text would be beneficial as it is currently only discussed briefly and pictured in figure 1 panel A, but seems like a large part of the novelty of the paper

L44 - consistency: the figure is related to the previous paragraph, I suggest moving the first sentence to the end of the previous paragraph

L49 - further information: could you elaborate on the term 'virtual dissection'? Describe this more fully either here or in the methods section, exactly what tools were used and how to carry it out. Maybe provide supplemental images of each step in the relevant software?

L47 - clarification: is the sample in the video an 84-year-old woman, but not in Figure 1? Figure 1 legend mentions a 60-year-old woman

L48 - further information/clarification: no signal-to-noise ratio is shown in Fig 1E, is there quantification to assist with the claims? Maybe incorrect language is used here. Quantification should be used to support the claim of homogeneous penetration of the light sheet

Page 3 (Figure 1)

L54 - clarification: it is not clear what is labelled in the figure panels without looking at the legend, potentially include the text 'Col IV' in the Panel D as in Panel K to indicate the labelling

L54 - clarification: for a naive viewer it is not clear what part of the eye is being imaged in each panel. Potentially indicate what part of the eye the viewer is seeing in each panel through text or other means (referring to panel B?).

Page 4

L63/64 - clarification: panel K does not appear to be from the blue dashed line, is it a different region, or a different eye?

Page 5 (Figure 2)

L84 - clarification: potentially include the name of the protein that is antibody labelled in each panel instead of only for multi-colour labels (or at panel A, and then panel E as it is different from the rest)

L84 - it is unclear what area of the eye is being shown, and in what orientation. Additional text could be included (same as comment on figure 1)

L84 - clarification: the scale of Figure 2 panels B and C have different scales, is this done intentionally? It may be better to show them at the same scale for comparative purposes

Page 6

L91 - clarification: are the scans of the same canal but in different orientations? Is the 3D rendering of the canal or the collector vein, the legend makes it unclear?

L100-102 - further information: elaborate on what these comparisons will mean to clinicians? What extra benefits are afforded by being able to make these comparisons to a cleared human eye post-mortem? It is currently unclear

L103-106 - further information: include these sample processing artifacts in pictures in a supplemental figure to highlight what users of this technique should look out for

L106 - further information: elaborate and suggest some refinements to the protocol?

Page 8

General methods comments:

- Include any safety concerns for hazardous chemicals
- Was everything done at room temperature unless otherwise stated? Were eyes kept in the fridge,

were chemicals kept in the fridge? etc.

- Is agitation mentioned at every step in which it was used?

- Will there be public sharing of the data for analysis? This is not mentioned.

L156 - further information: elaboration on what results to expect from the tissue at each step of the protocol (i.e. the eye should appear white or light pink after bleaching, assisted visually by figures in a supplement?)

L157 - further information: what apparatus/equipment was used for these incubations of large tissue? Is there a suitable material or volume of liquid for the vessel? Was there any agitation for the dehydration step?

L160 - further information: what kind of setup was used for agitation i.e. a rocker/shaker with Falcon 50 mL centrifuge tubes or some other equipment - this is beneficial for repeating the experiment and can affect penetration and bathing depending on the interpretation of the text.

L158 - clarification: what concentration of PBS was used at each step w/ methanol?

L166 - clarification: potentially a word missing after 'modulation of antibody-antigen' - is it binding?

L171 - clarification: anti-'Tubulin III'

L172 - clarification: are the antibodies incubated without the sample? Please clarify

L173 - further information: what equipment/apparatus with rotation was used, as RPM is mentioned but the apparatus is not defined.

L175 - clarification: is this an incubation of the antibody with PBSGT, without the sample? Or is this an incubation with the sample? Please clarify

L175/176 - further information: what secondary antibodies were used? Please provide product codes.

Page 9

L178 - clarification: is the 4% PFA in PBS or PBSGT

L185 - clarification: what is the imaging medium used? Is it DBE or something not mentioned?

L191 - clarification: what does the '42' indicate here?

L204 - clarification: this sentence does not make sense to me, why does the Z-stack 'respectively result in the in-plane pixel size'? I think this sentence should be two separate ones describing the axial and lateral pixel/voxel sizes if I have understood it correctly

L206-213 - further information: elaboration of these visualisation and analysis steps using figures would be beneficial to inexperienced users of the protocol

Supplemental video

- clarification: to assist the viewer in understanding what part of the eye they are seeing it could be useful to pause frames and label structures with arrows and text

T00:41 - it is not clear how virtual isolation is done, please include this wording in the methods so the user can refer to something to understand what is meant by this

Signed: Robert Lees

Answers to reviewers

We would like to thank the reviewers for their useful comments and advice.

We listed in a table a point-to-point answer to each reviewer's comment.

Reviewer 1

Referee #1: light sheet microscopy, structured illumination, image formation

Reviewer 1 wrote: In this manuscript, the authors describe the first full imaging of cleared human eyes using light sheet fluorescence microscopy and a customised iDISCO clearing protocol. The manuscript is clear and well written, with impressive results. I particularly enjoyed the supplemental movie, which presented an impressive demonstration of the main results.

Is it possible to obtain a scattering-only volume (cf. reflection confocal microscopy) to see all structures within the eye, not just those labelled with antibodies?	Lightsheet reflection microscopy for large volumes is not possible because of the nature of the illumination found in lightsheet microscopy. Indeed, in this microscopy modality, light has to penetrate inside the sample which is not compatible with reflective microscopy. However, it is still possible to image unlabelled brain structures. In iDISCO treated specimens, for example, the tissue autofluorescence in the 488 spectrum has often been used for a label free detection of inner structures and micro-structures. As the supplementary figure 2 shows, we can observe label free structures revealed by the autofluorescence in the 488 channel.
Is there a fundamental limitation that explains why only two-colour volumes are presented?	The Mesospim microscope at the Wyss Center of Geneva is able to image samples in four wavelengths: 405nm, 488 nm, 546 nm and 647 nm . The 405 nm is known for low penetration through the tissue whereas 488 nm is generally used for autofluorescence detection in samples treated with DCM and DBE. That is the reason why we privileged 546 nm and 647 nm excitation in this study.
However, I miss a fuller description of the imaging performance and comparison to other techniques, such as confocal microscopy. How much faster/easier is the light sheet imaging?	When compared to other techniques like confocal, lightsheet speed performances is 2–3 orders of magnitude faster than point-scanning methods. The lightsheet used in the work is based on the Axional Scanned Lightsheet microscopes (ALSM) technology and the scan speed is mainly dictated by the rolling shutter speed of the camera as well as the exposure time. We record at 5 FPS , whereas with a laser scanning confocal or a spinning disk we would record at 0.01-0.05 FPS or 0.1 FPS, respectively to have comparable readout.

	Large volumetric imaging using lightsheet comes with its own challenges, that are not only associated to the use of a custom-built microscope but also to the sample preparation and holding. This makes the process more complex than a standard specimens imaging with a confocal microscope. This has been integrated in the discussion part.
Are there regions of the sample that are less well resolved than others? What is the typical achieved resolution?	The image quality changes depending on the area of the sample. This is due to both, the large size of the sample and the moderate inhomogeneity of refractive index in different areas, which in turn makes the sample not fully optically transparent. These constraints were mitigated by using a dual side illumination to increase the penetration of the light through the whole sample, and by correcting the misalignment of illumination with the focal plane throughout the whole specimen thickness. The resolution depends on the NA of both illumination and detection objectives used for the experiments and ranges between 5-8 μm axially and laterally. This has been added in the discussion part, and we added a supplemental table with each image characteristics to precise them.
How much is the axial resolution improved by using the ASLM mode compared with a thicker, longer sheet?	Axially scanned light-sheet microscopy (ASLM) is an approach by which the “thinner part” of the focused laser beam (Rayleigh distance between 83 μm @ 4X to 284 μm @ 1X) is swapped across an entire field of view, ensuring the axial resolution is homogenous everywhere on the image. In ASLM mode, the mesoSPIM achieves an axial resolution of $6.52 \pm 0.07 \mu\text{m}$ (FWHM, $n=322$ beads, $nD=1.45$) across a FOV of 13.29mm (Voigt et al 2018 Nat. Methods). As an example, a Gaussian light-sheet with a 6.5 μm (1X magnification) waist would have a Rayleigh

	range of 284 μm. Enabling the ASLM mode thus corresponds to a 23.4-fold increase in FOV. On the other hand, longer lightsheet can be found in Bessel beams based lightsheet microscopy (lattice, etc ... 1 or 2 photons). However, in general, the length of the Bessel beam is typically on the order of tens of micrometers to a few hundred micrometers making these approaches incompatible with the size of the eyes that we have imaged in this study. This has been precised in the discussion part.
Furthermore, in Figure 1 panel C, the cleared eye has a distinct pink hue. Can the authors identify which residual component is responsible for this?	The pink/orange hue of the cleared eye is likely caused by saponin, used during the protocol for permeabilization, that is tinted orange/maroon.
It would appear that this may cause issues with illumination absorption near the outer boundary of the iris. Is this the case?	No issue with illumination absorption was observed around the iris.
The authors note that they remove the lens as they often observed reopacification of cleared lenses over time. Is any reopacification of other eye tissues observed?	In our experience with mouse eyes and human samples, no reopacification was observed elsewhere than the lens. Samples have now been cleared for a year, and all tissues are still cleared.
In the methods section, the authors state "this axially scanned light-sheet mode (ASLM) leads to a uniform axial resolution across the field-of-view (FOV) of 5 μm", but also "Z-stacks were acquired with a zoom set at 0.8X at 5 μm spacing". Hence, the Nyquist sampling criterion is not satisfied and so the true maximum possible axial resolution is 10 μm. Similarly, with an XY pixel pitch of 8.2 μm, this corresponds to a maximum possible resolution of 16.4 μm. What resolution	The reviewer is absolutely correct here. The effective detection NA of the MVX-10/MVPLAPO 1x combination varies with zoom and can be estimated as $\text{NA} = 0.065$ at 0.8\times (Olympus, personal communication). Based on these values, the diffraction-limited lateral resolution would be 3.8 μm. With 5 μm/pixel, the mesoSPIM is undersampled. The same argument applies for axial resolution. Here we privileged an isotropic voxel size on the actual best resolution. We made it clearer in the text.

is actually achieved in practice, and how does this vary over the sample?	
In the abstract, "light sheet fluorescence microscopy" is given the abbreviation "LFSM" which is mis-ordered. This choice is consistent throughout the manuscript, but would be less confusing if changed to the common LSFM.	This has been modified.
Figure 1 panel A misspells "fluorescence" as "fluorescente".	This has been done.
In the same panel, the timeline suggests that the light sheet imaging takes several months, which I do not think is the case (I believe the suggestion is supposed to be that the imaging can be started at any point within a window of several months before sample degradation). Perhaps the figure can be altered to make this less confusing.	This has been done.

Reviewer 2

Referee #2: inflammatory eye diseases, advanced imaging and bioinformatics

Reviewer 2 wrote: Authors Darche et al. report on a new method of optically clearing the human eye for light sheet fluorescence microscopy. The authors have demonstrated the advantage of this application in its ability to successfully visualise the ocular structures of the human eye in its native configuration to greater depth and higher resolution. This is an important area of study. However, there are several revisions that are recommended to strengthen the paper and make it accessible to a larger audience as it has the capacity to be of value to the wider field, outside of vision sciences.

--

Overall, I highly commend the authors on undertaking the series of experiments and obtaining the subsequent results that have led to the development of a new clearing and imaging method of the human eye. In particular, the ability to visualise the anterior segments of the eye, including the innervations is significant. There has no doubt that this project has required many optimisations and has met with numerous challenges and redirections. However, the persistence of this project to generate this data has enabled an important body of work to come to light, and become available to the scientific field. Therefore, the findings from this manuscript will be well received by the vision sciences, ocular immunology, neuroscience and other imaging fields. Well done.

Line 20 -23: references?	We added a citation about the diverse embryological origins of the several tissues of the eye to support this claim.
--

The introduction requires a more comprehensive literature review on the clearing and imaging methods that have been previously conducted on the human eye to date? If it is only very few, then there is space to indicate each one. In addition, what specifically makes this method better and the rationale for its use over others? This requires greater clarity in introduction. A search will demonstrate that other methods have been published (PMID: 28574496, PMID: 35859136). Together, this will help to support your statement in Line 32 – 33	In the article PMID: 28574496 a technique for clearing of parts of the eye without depigmentation, followed by transmission microscopy is described; only parts of the eye such as the sclera has been cleared, and there is no immunohistochemistry. Hence, we do not believe that it represents a work that is comparable. The paper PMID: 35859136 describes LSM of various organs, but not the eye. This paper has been added to the list of references. We integrated a more comprehensive literature review in the introduction and the references suggested by the reviewer.
Line 35 – explain linearization. Line 42- 43 – clarify what you mean by ‘modulation’ of antibody-antigen binding?	SDS is an anionic detergent, which allows coating of proteins in negative charges, leading to the loss of their tertiary structure and producing a linear protein molecule. Considering the link between antibody tertiary structure and antibody-antigen binding, it has been shown that any concentration of SDS over 0.01% leads to an inhibition of all immunochemical reactivity (Qualtiere et al., 1977). This property of SDS has been used in Murray et al., 2015, to develop their SWITCH protocol. In this, antibodies in contact with SDS will lose their antigen-binding properties, allowing them time to penetrate into the tissue without binding. Once the incubation time over, SDS is washed and the absence of SDS will lead to antibodies gaining back their antigen-binding properties, allowing for a more homogeneous staining, free from the usual gradient of strong labeling on the outside of the sample and weaker deep into the tissue.

	An inspired Switch protocol was also developed in our case to resolve the problem of the gelatinous human vitreous, whose meshwork tended to block antibodies penetration (an issue already encountered in flat mounted retinas). SDS played a role quite similar as the one used on proteins during Western Blot, allowing for easy passage through the mesh of the vitreous to easily reach the interior of the eye. We made it clearer in the discussion section.
Line 51 - 52 – not clear on how imaging of ocular vascular bed is achieved? Was vasculature stained with CD31?	All panels from figure 1 are from the same eye, of a 84-yo female, labeled with anti-collagen IV antibody, which strongly labels vessels. The different vascular beds from the eye were then isolated from the same 3D sample using manual segmentation in Imaris software, and isolated to be studied separately.
Line 44 – 53. The description of the figures in the results require more detail and thought. Although the figure legends do have detail within them, reference to the figures in the results should explain what procedure has been done and the appropriate structures that are being shown. Eg, what is the yellow arrows pointing to, scale bar on K?	We added details and thoughts about the results throughout the paper to answer the reviewer comments. Yellow arrows are described in the legend. Scale bar was added on panel K.
Line 51 – fig 1, F, G should be Fig F, G, I – please carefully check for formatting, grammatical errors and typos throughout	This has been read through by a native english speaker.

Upon immediate glance, the figures are not intuitive to what is being shown other than an assumption that it is blood vessels. To help the reader, it would be useful to have a small subtitle on each frame (Eg. Panel A – could have the title ClearEye Method), and if possible, to pseudo-colour the appropriate panels as was done in Figure K. Eg. It is not evident that Panel G vs H is of the vasculature of the retina and choroid, respectively.	Modifications to the figures have been made to be more intuitive to read.
Figure 1, Panel B – instead of number of the structures, can you annotate directly onto the eye schematic.	This has been done.
Same comments for Figure 2 as made for Figure 1	This has been done.
Line 70 – can anti-tubulin III be pseudo-coloured?	Sorry, we are not sure of the interpretation of the question?
Line 72 – Fig 2C – what are yellow arrows pointing to.	In 2C, the arrowheads are pointing to microneuromas in the cornea, labeled with anti-tubulin III labeling
Line 77 – explain why segmentation and 3D rendering is useful for the Schlemm’s canal	The Schlemm’s canal is a circular structure playing a crucial role in the aqueous humor circulation. As such, it can be postulated that its size and shape modulates the resistance to outflow. Accordingly, the morphology of the Schlemm’s canal is altered during glaucoma. This has been precised in the discussion.

Similar to comments made about introduction, here it would be useful to compare with other clearing and imaging systems that have been used and/or how the lack of such methodology as the one you have established has hampered the ability to fully visualize the delicate nerves, blood vessels and immune cells found within the eye.	We added a comparison of clearing techniques and imaging systems in the paper as suggested by the reviewer.
Line 151 – 153 – This begs the question of how important it is to fix tissues in 4% PFA O/N? Have you tried with different fixation times?	The fixation protocol has been determined as optimal through other projects in the team, and provided great conservation of samples through time, as well as easy handling for the eye bank in charge of sample collection. No other fixation time was tested for this paper.
Line 153 – fixed O/N at room temperature or 4 degrees C?	Eyes are fixed O/N at room temperature. This has been added to the manuscript.
Line 154- clarify whether removal of lens post-fixation doesn't lead to collapse of the eye.	No collapse of the eye was observed, since the lens in the human eye is a small structure (compared to the mouse eye). A small dent in the iris of one sample was the only consequence observed.
Line 159 – did the eyes have to be injected with hydrogen peroxide or were the eyes only immersed in bleaching solution?	Eyes were immersed in bleaching solution, with a gentle flow directed toward the interior of the eye using a pipette for each bleaching solution renewing. The sentence “To help with pigment elimination, gentle flow toward the interior of the eye is done using a pipette for each bleaching solution renewing .” has been added in the text.
Line 162 – approximately how many PBS washes across the day and how long was each?	Baths are around 90 minutes, and four to six baths are made during the day to ensure complete washing of the sample

Line 166 – do you mean antibody-antigen binding?

Line 166 – what specifically is SDS doing, explain? Does it act like antigen-retrieval to enhance epitope binding?

SDS is an anionic detergent, which allows coating of proteins in negative charges, leading to the loss of their tertiary structure and producing a linear protein molecule. Considering the link between antibody tertiary structure and antibody-antigen binding, it has been shown that any concentration of SDS over 0.01% leads to an inhibition of all immunochemical reactivity (Qualtiere et al., 1977).

This property of SDS has been used in Murray et al., 2015, to develop their SWITCH protocol. In this, antibodies in contact with SDS will lose their antigen-binding properties, allowing them time to penetrate into the tissue without binding. Once the incubation time over, SDS is washed and the absence of SDS will lead to antibodies gaining back their antigen-binding properties, allowing for a more homogeneous staining, free from the usual gradient of strong labeling on the outside of the sample and weaker deep into the tissue.

An inspired Switch protocol was also developed in our case to resolve the problem of the gelatinous human vitreous, whose meshwork tended to block antibodies penetration (an issue already encountered in flat mounted retinas). SDS played a role quite similar as the one used on protein during Western Blot, allowing for easy passage through the mesh of the vitreous to easily reach the interior of the eye.

Line 172 – antibodies incubated in PBSGT-Sw for several hours – how many hours is that?	Antibodies were incubated around two hours before being used on samples. This has been added to the text for clarity.
Line 180 – 182 – explain why you decided to use DCM and DBE? Are these used in other published clearing papers? If yes, reference.	Our expertise is in iDISCO+ derived protocols, since DCM and DBE have been used by the team for previous projects on the mouse eye (Darche and al, communication biology, 2022) and proved very efficient in clearing mouse eyes.
Line 183 – how long and under what conditions can cleared samples be stored in DBE for? Is this the final storage medium?	Samples are stored in DBE in dark glass contained, at room temperature. No precise time of conservation has been determined yet, since a year after clearing samples are still clear and labeling strong. DBE is the final storage medium This has been added in the methods.
Line 185 – what is the imaging medium?	The imaging medium is DBE. This has been added to the text.

Reviewer 3

Referee #3: light sheet microscopy

Reviewer 3 wrote:

Brief summary of the manuscript

=====

This manuscript is a brief communication that, for the first time, proves the utility of sample clearing and light sheet fluorescence microscopy for imaging intact whole human eyes. The authors have developed a protocol for clearing and imaging human eyes, ClearEye, based on their previous experience working with mouse eyes. The authors briefly discuss the relevance of the resulting images and analysis techniques for clinical research.

Overall impression of the work

The manuscript is technically sound using adaptations to well-known techniques and the concept is a logical extension from previous methods of clearing mouse eyes, <https://www.nature.com/articles/s42003-022-04104-2>. The work is convincing, however there could be further effort to describe the methodology and analysis to aid readers in repeating this work. Additionally, refinement of the technique to be reliable would be beneficial to the community, or at least highlighting the critical steps that are unreliable.

This paper will influence thinking in the field, as it opens clinical researchers to the option of imaging intact whole human eyes. The exciting features of this work are the novel images of whole human eyes at a decent resolution (8 um), the outstanding ease with which many structures in the eye can be identified, and the data manipulation possible using virtual dissection. The workflow is not necessarily relevant to those outside of the field, as it is specific to whole human eyes.

No other experiments would strengthen this paper in my opinion, the experiments conducted are of utility on their own and appear novel. The methodology is completely manual, no new tools for virtual dissection are used or created. There appears to be no major biological claims from the paper, so this is purely a technical paper, using examples of the biology of the eye.

I congratulate the authors on their efforts!

Signed: Robert Lees

As one of the main outcomes of the paper is the description of the clearing protocol, it should be described in more detail by: - Including an extra figure (main or supplemental) to describe the sample processing steps in more detail (images of the setup, apparatus etc.), expand on methods section slightly	We implemented a more comprehensive description of all methods steps and protocol development choices, and detailed as much as possible the methods section for better reproducibility.
Including an extra figure (main or supplemental) to describe the analysis conducted, expand on methods	This has been added to the paper in an extra figure and expanded methods
If this is an unfinished protocol with issues (see line 106), it may be best to make the refinements before publication	We modified the sentence, which was not well formulated.

Further discussion of the context and benefit of the technique: - Over existing techniques - For clinicians or ophthalmology research in general	We developed these ideas more in the discussion and introduction of the paper, as suggested.
Grammatical error corrections to improve readability	The text has been read by a native English speaker to improve the readability.
Copy-editing would benefit this manuscript to improve grammar and therefore readability as Nature Communications Biology does not undertake copy-editing for authors, examples include L16 grammar, L23 grammar, L31/32 repetition, L46/47, L55, L57, L58, L78, L97, L100-102, L111, L173	The text has been read by a native English speaker to improve the readability.
Headings for the text sections would be highly beneficial to allow the reader to navigate the paper	This has been done.
The development of an analysis tool specific to data from the eye would be beneficial and should be discussed, as currently it is done by hand with non-specific tools	It has been added in the discussion section.
Are there any novel biological findings that can be highlighted?	Our technique demonstrates that LFSM of human eyes will allow better characterization of structures for which data in diseased patients are still relatively scarce such as the Schlemm's canal, the nerves of the cornea, the circulation in the iris and optic nerve. As far as we know, we are the first to present the entire 3D human retina/choroidal vascularization, without any tissue dissection or cutting.
Discuss the comparison of this protocol to the existing best methods (sectioning and mounting) using a figure, discuss or show resolution differences and discuss what can be resolved using this technique over others	This discussion comparing LFSM of entire human eyes compared to traditional best methods in histology has been added in the discussion part.
Line 13 - grammar: should LFSM be LFSM for Light sheet fluorescence microscopy? Written throughout paper as LFSM	This has been modified

L20-23 - further information: is there a reference for this claim?	A reference has been added in the introduction.
L29 - inclusion of this reference may be beneficial here: https://www.ncbi.nlm.nih.gov/pmc/articles/PMC8374524/	We thank the reviewer for this comment, this has been added to the text.
L35 - further information: could you elaborate on linearisation further somewhere in the methods or text as it is not clear from the references what this means	SDS is an anionic detergent, which allows coating of proteins in negative charges, leading to the loss of their tertiary structure and producing a linear protein molecule. Considering the link between antibody tertiary structure and antibody-antigen binding, it has been shown that any concentration of SDS over 0.01% lead to an inhibition of all immunochemical reactivity (Qualtiere et al., 1977). This property of SDS has been used in Murray et al., 2015, to develop their SWITCH protocol. In this, antibodies in contact with SDS will lose their antigen-binding properties, allowing them time to penetrate into the tissue without binding. Once the incubation time over, SDS is washed and the absence of SDS will lead to antibodies gaining back their antigen-binding properties, allowing for a more homogeneous staining, free from the usual gradient of strong labeling on the outside of the sample and weaker deep into the tissue. An inspired Switch protocol was also developed in our case to resolve the problem of the gelatinous human vitreous, whose meshwork tended to block antibodies penetration (an issue already encountered in flat mounted retinas). SDS played a role quite similar as the one used on protein during Western Blot, allowing for easy passage through the mesh of the vitreous to easily reach the interior of the eye. We made it clearer in the discussion.
L36 - clarification: what is 'extensive numerical dissection', does this mean 'analysis'? A more informative explanation for the analysis could be used here	“Extensive numerical dissection” refers to manual segmentation of different ocular structures. The Imaris software tool “segmentation” allows for plane by plane segmentation, to create a 3D segmentation through the entire sample, allowing to isolate in silico different tissues to study them separately. This has been described better in the

	result section and a supplementary figure has been added to illustrate the necessary steps.
L39 - further information: if this is an adaptation of the mouse eye clearing protocol, a comparison to that protocol would be beneficial here to highlight what was changed and why the human eye required the adaptation (the current description is not specific to the 'human' eye)	If the clearing protocol for the mouse eye and the ClearEye protocol for humans present some similarities, all steps present very different timing (longer for the human samples), and with differences in labeling protocol. The mouse eye has a very liquid vitreous, that does not hinder antibody penetration, less melanin, and much thinner/frail tissues. The human eye is highly pigmented, with a very dense vitreous known to hinder labeling even in small quantities on flatmounted retinas, and thick sclera. Hence, the protocol for human eyes has to be stronger, while still avoiding any damage to the thin and frail retina. The differences have been discussed in the discussion section and a schematic of the key steps of the ClearEye protocol added as a supplementary figure.
L39 - further information: the steps taken to develop the protocol are not shown and may be beneficial to help the community understand decisions made when clearing different tissues. Include data from the development of the protocol if possible? Was a reference used to assist your choices, or was it trial and error? Some elaboration of the clearing protocol in the main results text would be beneficial as it is currently only discussed briefly and pictured in figure 1 panel A, but seems like a large part of the novelty of the paper	The protocol has first been developed on pig eyes, due to their similarities to the human eye (size, pigmentation, tissues and vitreous properties). Due to the absence of a MesoSpim in France, samples underwent different variations of the protocol before dissection to image the different tissues using epifluorescent and confocal microscopy, to evaluate immunolabeling through every tissue. The protocol was based on solid references (iDISCO+ is used in our institute and we published on its use on the mouse eye; Murray et al., published the SWITCH protocol dedicated to homogeneous antibody labeling through brain tissue) but the complete protocol for the human eye was made through trial and errors, based on knowledge of the physical properties of all ocular tissues. Elaborations on the protocol have been added in the text on your advice and discussed in the discussion section.
L44 - consistency: the figure is related to the previous paragraph, I suggest moving the first sentence to the end of the previous paragraph	This has been done.

L49 - further information: could you elaborate on the term 'virtual dissection'? Describe this more fully either here or in the methods section, exactly what tools were used and how to carry it out. Maybe provide supplemental images of each step in the relevant software?	“Extensive numerical dissection” refers to manual segmentation of different ocular structures. The Imaris software tool “segmentation” allows for plane by plane segmentation, to create a 3D segmentation through the entire sample, allowing to isolate in silico different tissues to study them separately. A supplementary figure on this 3D segmentation has been added to the paper.
L47 - clarification: is the sample in the video an 84-year-old woman, but not in Figure 1? Figure 1 legend mentions a 60-year-old woman	This has been modified in the text, the sample was indeed from a 84 yo woman.
L48 - further information/clarification: no signal-to-noise ratio is shown in Fig 1E, is there quantification to assist with the claims? Maybe incorrect language is used here. Quantification should be used to support the claim of homogeneous penetration of the light sheet	No quantification of the signal-to-noise ratio was realized. The wording in the sentence has been modified.
L54 - clarification: it is not clear what is labelled in the figure panels without looking at the legend, potentially include the text 'Col IV' in the Panel D as in Panel K to indicate the labelling	This has been modified in all panels for better clarity.
L54 - clarification: for a naive viewer it is not clear what part of the eye is being imaged in each panel. Potentially indicate what part of the eye the viewer is seeing in each panel through text or other means (referring to panel B?).	This has been implemented on each panel, for better understanding for naive viewers.
L63/64 - clarification: panel K does not appear to be from the blue dashed line, is it a different region, or a different eye?	Panel K is from a different region, only the J panel (in blue) is localized on the blue dashed line. Clarification has been made to the legend.
L84 - clarification: potentially include the name of the protein that is antibody labelled in each panel instead of only for multi-colour labels (or at panel A, and then panel E as it is different from the rest)	This has been added in each panel.
L84 - it is unclear what area of the eye is being shown, and in what orientation.	This has been added in each panel.

Additional text could be included(same as comment on figure 1)	
L84 - clarification: the scale of Figure 2 panels B and C have different scales, is this done intentionally? It may be better to show them at the same scale for comparative purposes	Yes, this is done intentionally. Panel B highlights the nerves in the cornea at a larger scale, while panel C focuses on small abnormalities in the nerves. Details of microneuromas would be lost at a lower magnification while a close up of the nerves in panel B would not allow for visualization of the network of nerves in the cornea.
L91 - clarification: are the scans of the same canal but in different orientations? Is the 3D rendering of the canal or the collector vein, the legend makes it unclear?	Scans in E and F are of the schlemm's canals of two different donors. Both are imaged at a different axis (90 degrees difference) to show different angles of analysis of the same structures through different samples. 3D rendering is of the Schlemm's canal, and has been added directly on the panel for more clarity.
L100-102 - further information: elaborate on what these comparisons will mean to clinicians? What extra benefits are afforded by being able to make these comparisons to a cleared human eye post-mortem? It is currently unclear	Clinical examination of the fundus of the eye is done by both en face imaging (e.g. funduscopy) and transverse imaging (OCT). The current gap between histology and in vivo imaging arises from the fact that 3D reconstruction and en face viewing using conventional sectional histology is cumbersome and by essence prone to reconstruction artifacts inherent to realignment. Furthermore, sectioning is done in parallel sections which cannot be done perpendicularly to the RPE for all sections. Confocal microscopy allows 3D reconstruction and customized displays, but has a relatively limited field of view compared to LSFM, and is an order of magnitude more time-consuming. Therefore, LSFM of intact eyes will facilitate the comparison of in vivo imaging to anatomical structures. Indeed, customized displays reproducing all in vivo imaging modalities, including a "funduscopy-like" can be done. Hence, LSFM of cleared eyes is therefore more convenient for the exploration of the phenotypic spectrum of human diseases by histology. Scanning through the entire sample will facilitate detection of focal lesions. Extensive documentation of diseases affecting the entire eye such as myopia, glaucoma and uveitis will be possible.

L103-106 - further information: include these sample processing artifacts in pictures in a supplemental figure to highlight what users of this technique should look out for	This has been done
L106 - further information: elaborate and suggest some refinements to the protocol?	At the moment, no alteration to the protocol has been considered, since the few artifacts have not been an issue in our studies.
Methods: Include any safety concerns for hazardous chemicals	All Safety and concerns for chemicals used in this protocol can be found in the products information online.
Was everything done at room temperature unless otherwise stated? Were eyes kept in the fridge, were chemicals kept in the fridge? etc.	Everything was done at room temperature unless otherwise stated. Fixated eyes were kept in the fridge at 4°C before experiment, then in glass containers filled with DBE after clearing. Chemicals are all kept in ventilated cabinets, except for pure H2O2, which is kept at 4°C. This has been added to the text.
Is agitation mentioned at every step in which it was used?	This has been verified and added as needed in the text
Will there be public sharing of the data for analysis? This is not mentioned.	Complementary images not included in the paper will be uploaded on a dedicated website. Raw data will be shared for collaborations on demand.
L156 - further information: elaboration on what results to expect from the tissue at each step of the protocol (i.e. the eye should appear white or light pink after bleaching, assisted visually by figures in a supplement?)	A detailed protocol with pictures of each critical step will be made available on our website or on demand.
L157 - further information: what apparatus/equipment was used for these incubations of large tissue? Is there a suitable material or volume of liquid for the vessel? Was there any agitation for the dehydration step?	Samples were incubated in large plastic containers for bleaching/ labeling steps and in glass containers for clearing. Minimal volume used was of 60mL for the antibodies mix, all other incubations were made in a minimal volume of 100 mL. Dehydration step was made under gentle agitation. This has been added to the text
L160 - further information: what kind of setup was used for agitation i.e. a rocker/shaker with Falcon 50 mL centrifuge tubes or some other equipment - this is beneficial for repeating the experiment and can affect penetration and	Two type of agitation are used through the protocol:  - During labeling, when samples are incubated at 37°C, agitation is of 70 RPM by rotation of the platform in the mini Incu-shaker mini from Benchmark scientific.

bathing depending on the interpretation of the text.	- For other steps at room temperature, samples were on a rocker, at Ms Major Science Rocking Shaker, on a low rocking setting. These precisions have been added to the text.
L158 - clarification: what concentration of PBS was used at each step w/ methanol?	PBS 1X was used.
L166 - clarification: potentially a word missing after 'modulation of antibody-antigen' - is it binding?	This has been modified.
L171 - clarification: anti-'Tubulin III'	This has been modified
L172 - clarification: are the antibodies incubated without the sample? Please clarify	Antibodies are first incubated for at least two hours without the samples, to ensure a complete reaction with the SDS present in the PBSGT-Sw solution before contact with the sample.
L173 - further information: what equipment/apparatus with rotation was used, as RPM is mentioned but the apparatus is not defined.	The apparatus used is Incu-shaker mini from Benchmark scientific. This has been added to the methods.
L175 - clarification: is this an incubation of the antibody with PBSGT, without the sample? Or is this an incubation with the sample? Please clarify	This is an incubation of the sample only in PBSGT, without any supplementary antibodies. The absence of SDS in the medium allows for Antibodies already inside the tissue to bind to their antigen.
L175/176 - further information: what secondary antibodies were used? Please provide product codes.	A comprehensive supplementary tables with all informations on antibodies, dilutions, and imaging characteristics has been added.
L178 - clarification: is the 4% PFA in PBS or PBSGT	PFA is always diluted in PBS1X in this protocol
L185 - clarification: what is the imaging medium used? Is it DBE or something not mentioned?	The imaging medium is DBE. This has been added to the text.
L191 - clarification: what does the '42' indicate here?	42 is a typo, deleted in the text
L204 - clarification: this sentence does not make sense to me, why does the Z-stack 'respectively result in the in-plane pixel size'? I think this sentence should be two separate ones describing the axial and lateral pixel/voxel sizes if I have understood it correctly	Z-stacks were acquired with a zoom set at 0.8X and resulting in 2048x2048 pixels frames at 8.2 X 8.2 X 5 μ m voxel size. We added a supplementary table to precise imaging characteristics.
L206-213 - further information: elaboration of these visualisation and analysis steps using figures would be	This has been detailed in a supplementary figure (Supplementary figure 1)

beneficial to inexperienced users of the protocol	
Supplemental video - clarification: to assist the viewer in understanding what part of the eye they are seeing it could be useful to pause frames and label structures with arrows and text	This has been done
T00:41 - it is not clear how virtual isolation is done, please include this wording in the methods so the user can refer to something to understand what is meant by this	The virtual isolation is done manually. This has been added in the text and as a supplementary figure.

REVIEWERS' COMMENTS:

Reviewer #1 (Remarks to the Author):

Thanks to the authors for the revisions made to their manuscript — this is now much clearer and I am happy to support publication in Communications Biology.

Reviewer #2 (Remarks to the Author):

Fine from my end for submission. All major concerns addressed. Urge authors to do full review of article and double check for spelling mistakes and grammatical errors.

Reviewer #3 (Remarks to the Author):

The authors have done a thorough job in replying to any concerns. The manuscript is now more informative, which helps with reproducing the results, as well as interpreting the images. I congratulate the authors on their work, well done!

Signed: Robert Lees